# Metacognitive Interpersonal Therapy for Misophonia: A Single-Case Study

**DOI:** 10.3390/brainsci14070717

**Published:** 2024-07-17

**Authors:** Eleonora Natalini, Alessandra Fioretti, Rebecca Eibenstein, Alberto Eibenstein

**Affiliations:** 1Tinnitus Center, European Hospital, Via Portuense 700, 00149 Rome, Italy; eleonora.natalini@gmail.com (E.N.); rebecca.eibenstein95@gmail.com (R.E.); 2Department of Biotechnological and Applied Clinical Sciences, University of L’Aquila, 67100 L’Aquila, Italy; alberto.eibenstein@univaq.it

**Keywords:** misophonia, maladaptive schema, Metacognitive Interpersonal Therapy, personality disorder, hearing disorder

## Abstract

Background: Misophonia is a chronic condition in which the exposure to specific sounds increases the arousal and recurrence of specific intense negative emotions. We hypothesized that misophonia may be strongly related to maladaptive interpersonal schemas that create difficulties in interpersonal relationships. Subjects with maladaptive interpersonal schemas think that other people try to subjugate, criticize, dominate, exploit, deceive, disregard, and humiliate them. Furthermore, these patients typically endorse a representation of self as mistreated, constricted, harmed, damaged, humiliated, impotent, inadequate, or fragile. Methods: We describe the course of a treatment of Metacognitive Interpersonal Therapy (MIT) in a young man presenting misophonia and co-occurrent obsessive–compulsive personality disorder (OCPD) and avoidant personality disorder (AvPD), with narcissistic traits and normal hearing. We collected qualitative and quantitative data at the beginning of the intervention and at 2 years follow-up. Results: The therapy aimed at increasing awareness of maladaptive interpersonal schemas and promoting a healthy self. The results reported a significant decrease in misophonia; behavioural experiments were used to increase the quality of social relationships and tolerance to the trigger sounds. Conclusions: MIT can be an effective therapy for the treatment of misophonia.

## 1. Introduction

Misophonia is a disorder of decreased tolerance to specific sounds or stimuli (triggers) associated with such sounds, as reported by the consensus definition of Swedo et al. [1]. Specific sounds and/or related sensory inputs produce strong negative emotional and behavioral reactions (anger, disgust, and frustration) not typically observed in the general population. The expression of misophonic symptoms is typically first observed in childhood or early adolescence. The estimated prevalence of misophonia is about 8% to 20% [2]. Some authors suggest that misophonia may be classified as a psychiatric disorder [3,4]. Some psychiatric disorders, like anxiety, depression, and obsessive compulsive-related disorders, may be represented as a comorbidity of misophonia [5]. Auditory disorders like hearing loss, tinnitus, recruitment, phonophobia, and hyperacusis must be evaluated in patients with misophonia. Tinnitus is a phantom auditory perception and, based on the neurophysiological model of Jastreboff, the neuronal networks involved in tinnitus and misophonia are identical [6]. The auditory system is needed for the perception of tinnitus and misophonia, but the limbic and autonomic nervous systems are the main systems responsible for negative tinnitus- and misophonia-evoked reactions. Conditioned reflexes explain why there are problems with tinnitus or misophonic triggers. There are two paths in the network processing of tinnitus signals and activity evoked by bothersome sounds: a conscious path, which involves cognitive processing of the signal, and which is dominant at the initial stages of tinnitus or misophonia; and a subconscious path governed by the principles of conditioned reflexes, which appears to be dominant in chronic tinnitus or misophonia. Hyperacusis is defined as a decreased sound tolerance described as a “physical discomfort or pain” in response to sound levels typically tolerable in the general population [7]. Tyler and al. described four subtypes of hyperacusis: loudness, annoyance, fear, and pain [8]. Case history, questionnaires, psychiatric and/or psychological evaluation of comorbid psychiatric disorders, and audiologic evaluation of comorbid auditory disorders are strongly suggested to determine diagnostic and therapeutic efficacy [9].

Research and health care regarding misophonia are currently scarce. Specific studies on the treatment of misophonia are also limited. The most widely used therapeutic approach is Cognitive-Behavioural Therapy (CBT) [10]. 

Basic CBT techniques, such as cognitive restructuring and the use of functional behavioural coping strategies, are frequently used tools [11,12,13,14]. Cognitive restructuring is used to modify patients’ distorted thoughts, e.g., “he makes this sound voluntarily to annoy me”, while behavioural strategies are mainly used to contrast the avoidance tendency. 

In addition, relaxation techniques and task concentration exercises are useful in misophonia to manage emotions, especially anger [15,16,17]. The article by Gregory and Foster [18] describes in detail the five sessions (12 h in total) of CBT treatment of misophonia on a young patient. The work focuses on the patient’s difficulties in expressing her needs, her view of others as disrespectful and uncaring, and her perception of herself as wrong and problematic. Similar observations emerged in the case series by Gregory et al. [19]. Some works use third-wave approaches. In the study by Jager and colleagues [20], Eye Movement Desensitization and Reprocessing (EMDR) is used on 10 patients to process memories associated with early misophonia experiences in order to reduce symptoms and improve the quality of life. The article by Kamody and Del Conte [21] reports the use of Dialectical Behaviour Therapy (DBT) in a young patient. The treatment focuses on anger management with an improvement of extreme to moderate misophonia. The patient was also given drug treatment. Mindfulness and the components of Acceptance and Commitment Therapy (ACT) and DBT were used in the treatment of a young adult [22]. The work focused on the development of acceptance, non-judgement, and mindfulness. The strategy of opposite action was used to counter dysfunctional behaviours related to misophonia. The application of CBT techniques enhanced with elements of ACT, DBT, and Mindfulness was carried out online with a 16-year-old girl. The 15 week treatment focusing on emotional regulation resulted in an improvement of misophonia from severe to mild [23].

Metacognitive Interpersonal Therapy (MIT) belongs to third-wave therapies and is an integrated approach mainly used for the treatment of personality disorders [24,25,26]. 

The therapy aims to modify the patient’s maladaptive interpersonal schemas. Interpersonal schemas indicate how the patients see themselves and others [25,27,28]. When a need or desire is activated, people predict how others will react to their demands. Typically, when maladaptive schemas are active, patients see others as critical, judgmental, rejecting, indifferent, or dominant. At the same time, they view themselves as wrong, fragile, vulnerable, powerless, or submissive. 

The intentions that people attribute to others direct their emotions, thoughts, and actions. For example, if I think a person may criticize me for something I want to do, I may feel shame and sadness, think I am worthless, and withdraw before I even begin any action. Often the same dysfunctional patterns hinder patients in achieving their goals. Starting with specific autobiographical episodes, a shared formulation of the patient’s functioning is constructed with the patient in order to make him/her aware of his/her own patterns. MIT, over the years, has integrated many experiential techniques, including guided imagery and rescripting, role-play and two-chair approaches, bodily exercises, and behavioural experiments [26,29]. 

Our hypothesis is that in patients with misophonia, these maladaptive interpersonal schemas underlie and sustain the disorder [30,31]. Indeed, patients with misophonia often think that the actions of others are intentional and disrespectful of their needs [11,13,18,30]. The onset of misophonia also depends on the context or source of the sound [30,32]. In patients with misophonia, the view of self is impaired. They may feel wrong, powerless, or guilty [18,23]. The presence of maladaptive interpersonal patterns can lead to one or more personality disorders. The purpose of our study is to detail the MIT treatment of a patient with misophonia in order to highlight the importance of interpersonal schemas in maintaining the disorder. To our knowledge, this is the first study of misophonia successfully treated with MIT. 

## 2. Materials and Methods

### 2.1. Presenting Problem

Marco is a 25-year-old man with symptoms of misophonia since the age of 10. In the early years, however, certain sounds were rarely perceived as threatening and they did not create great discomfort. 

The problems with misophonia became stronger after high school, at the age of 18, when he spent more time at home, went to university, and started his first job. During university, which he did not complete, he had difficulties with his roommate’s breathing, which he felt was too heavy. While studying, he was annoyed by the music he heard coming from his earphones. 

During his first job, working and living at home with the same people, the trigger sounds were those related to swallowing and chewing, the same ones he was currently complaining of in his family. The difficulties with sounds were less with friends or strangers. 

Marco used earphones. Initially, the volume was low, then increased to the point where he could no longer hear the family’s conversations at the table. When he got up from the table, his anger was so intense that he often vented it by slamming doors or kicking objects.

Marco presented overcontrol and rigidity that were also visible in his posture. His life was characterized by self-imposed habits and rules. He set the alarm clock early every day even if he had no commitments, his diet was very controlled by excluding certain foods (e.g., carbohydrates because they make him drowsy), and he had significant difficulty with spending money. There is little space for spontaneous desire and no awareness of the difference between what he wanted and what was right to do or what he had to do. The tendency to perfectionism is a coping strategy that emerges from the narratives and the fear of making mistakes and of not doing everything as it should be, which hampers the paths he takes. Faced with the possibility of a difficulty, he generally tended towards avoidance. As previously reported, Marco has often changed study or work paths. He worked in a company in the administration office and sometimes helped his father in one of the family businesses. He joined a group of close friends and has never had an intimate relationship. 

Marco came to the Tinnitus Center in Rome, in July 2021, having previously tried two other therapeutic paths for the problem.

### 2.2. Assessment and Instruments

The diagnosis of misophonia was made by a clinical interview based on the criteria described by Schröder and colleagues and with the Amsterdam Misophonia Scale (A-MISO-S) [3]. A structured clinical interview for DSM-IV personality disorders (SCID-II) [33], thet Beck Depression Inventory-II (BDI-II) [34], and the State-Trait Anxiety Inventory (form-Y) (STAI-Y) [35] were also administered at the beginning of treatment.

The A-MISO-S is a semi-structured interview that is not validated. On a six-item scale (range 0–24), patients were asked about the (1) time they spend on misophonia, (2) interference with social functioning, (3) level of anger, (4) resistance against the impulse, (5) control they had over their thoughts and anger, and (6) time they spend avoiding misophonic situations. Scores from 0 to 4 are considered subclinical misophonic symptoms, 5–9 mild, 10–14 moderate, 15–19 severe, and 20–24 extreme. 

The SCID-II is a structured clinical interview that assesses the full range of PD traits found in DSM IV PD. The interview was administered by the treating clinician before the beginning of any physical or psychological treatment.

The BDI-II is a 21-item measure assessing depression over the previous 2 weeks. Higher scores suggest a high level of depression. The cutoff used are the following: 0–13 corresponded to minimal depression, 14–19 to mild depression, 20–28 to moderate depression and 29–63 to severe depression.

The STAI-Y is a self-report instrument measuring state-anxiety (anxiety about an event) and trait-anxiety (anxiety level as a stable characteristic). All items were rated on a 4-point Likert Scores range from 20 to 80. Higher scores have a correlation with a higher level of anxiety.

The score obtained from the A-MISO-S by the patient at the beginning of the treatment was 16, a severe grade. The SCID-II showed the presence of obsessive–compulsive personality disorder (OCPD) and avoidant personality disorder (AvPD), with narcissistic traits. The BDI-II reported a minimal depression (score 12) and the STAI-Y a medium-low level of both state-anxiety (score 46) and trait-anxiety (score 49). Normal hearing was defined with a hearing threshold < 25 dB HL in all tested frequencies (0.25–8 kHz) in both ears at the audiometric evaluation. Hyperacusis was excluded by testing the loudness discomfort levels (LDLs) at the frequencies of 0.25, 0.5, 1, 2, 4, and 8 kHz. The patient had normal hearing and normal LDLs.

### 2.3. Course of Treatment

In the first year, Marco’s therapy was carried out in person on a weekly basis. There was a break of about 6 months requested by the patient himself who was abroad for work reasons. Subsequently, therapy was resumed in online mode. Currently, sessions aree held every fortnight. 

The treatment was carried out by a psychotherapist with MIT training and work experience of about 10 years, 8 of which in hearing disorders.

#### 2.3.1. Shared Formulation of Functioning

As indicated by MIT, the first part of the treatment focused on the construction of a shared formulation of the patient’s functioning. In the first meetings, exercises of exploration and observation of the internal state were assigned to improve narrative and self-reflective skills.

Marco was asked to remove the earphones at trigger moments related to misophonia to check which thoughts, emotions, physical sensations, and action tendencies he was experiencing. Attentional techniques to be used in case of excessive discomfort were described and explained. Marco was already familiar with mindfulness, and this helped the process. In addition, he was given descriptive emotion cards to help the patient name his feelings.

The first part of the treatment was used to collect narrative episodes, i.e., detailed autobiographical memories that are well located in time and space. The focus was then placed on the details of these episodes to search for feelings, ideas, and motivations for actions. The observation and narration of the episodes led to the identification of several interesting points. Anger, typical of misophonia, emerged because Marco felt forced by others not only to witness unpleasant sounds but because others ate unhealthy food. This last observation was related to the feeling of disgust, of a moral kind, which also emerged in another observation. Marco felt more misophonia-related distress towards people who did not help tidy up the table and whom he judged to be rude, with no respect for others.

The exercises also helped to identify problems within the family. At the table, quarrels and discontent were the order of the day, and Marco suffered especially due to his mother, who was described as complaining and discontented. His mother often refused to eat with them when she was angry and this aroused feelings of anger and guilt in Marco. In the past, there was aggressive behaviour when the parents argued, with the father breaking objects and the mother damaging her own person as a demonstrative act of her suffering.

On a bodily level, Marco’s physical sensations were characterized by stomach tension and muscular rigidity that could result in aggressive behaviour that Marco later regretted.

Marco began with the therapist to see how misophonia was the only way to express anger because, from his point of view, his complaints were correct and the others were in the wrong. 

After analysing the misophonia episodes, he was asked to bring back other episodes related to the emotion of anger in therapy to see what interpersonal patterns might be activated. The investigation was not easy because, outside of misophonia, for the reasons described above, it was difficult for Marco to tell himself that he was feeling that emotion.

It was possible to identify moments of strong stress related to his father’s requests for help at work. Marco did not want often to help him but did so because he would otherwise feel selfish. The feeling of guilt also came back when his mother showed signs of impatience or sadness.

A healthy desire to live alone emerged during the interviews. The family owned a house that was sometimes used by the father because it was closer to work and because of his bad relationship with his wife. Some meetings were dedicated to the possibility of using this house, but Marco found it very difficult to ask—he felt he did not deserve this possibility. During the session, the therapist led Marco to evoke autobiographical memories related to the current situation in order to show the patient the existence of recurrences in the way of relating.

T: Does this belief that you don’t deserve things remind you of something from the past? Where could you have heard it?

P: When I was a child, I never had fashionable clothes and I was very ashamed. My mother wouldn’t buy them for me not to spend money. I was angry but then I stopped asking… I don’t know then at some point it felt right. Once, however, I had asked for a game and she bought it for me… I wanted it so much, I was happy, but then my mother… (she turns dark)

T: What’s happening? I can see sadness on your face…

P: Yes… I remember that I saw her suffering… it wasn’t a gesture made in joy, I felt that it was wrong, that the shopping was damaging the family, that I was a bad child… instead I had to be a good child, I didn’t have to cause problems.

T: And how did you feel towards your mother?

P: I felt guilty. Both she and dad always said they stayed together for us children and did things for us. I felt… I don’t know… ungrateful, if I only thought of myself. It’s dawning on me that my mother also used to complain a lot when she had to take me to football school, she made me feel it was weighing on me…

The information obtained through the narrative episodes and autobiographical memories allowed the reconstruction of interpersonal schemas. The desire for autonomy was hindered by the perception of himself as selfish and ungrateful if he took care of his own needs, and he felt guilt for this. The perception of the other was of a person who could be hurt by his desires or needs. In this case he felt the relationship threatened. Sometimes, however, the feeling of anger was present when he perceived the other as controlling and domineering. In this case, Marco felt that the other wanted to subjugate him, and his self-image as a person capable and deserving of autonomy was activated. In this case the healthy part emerged, as it is called in MIT. Perceiving the other as dominant also activated the motivational schema of social rank. This emerged to counter the self-image, not only as inferior because of submissiveness, but also inadequate and incapable. High morality and the need to follow certain rules were used in order to perceive oneself as better than others and wiser. The dysfunctional coping strategies related to the schemas were perfectionism, the use of high morality, and avoidance.

The reconstruction of the patterns was accepted by Marco with great interest and openness. Marco agreed with the reported reflections, and this allowed the transition to the next phase of therapy concerning change.

#### 2.3.2. Change Promoting

At the change promoting stage, clients are helped to take a critical distance from their schemas to build new ways of thinking and feeling in order to implement more adaptive behaviours.

A first step towards differentiation, that is, taking a critical distance from one’s schemas, was obtained with the use of mindfulness and guided imagination to remind Marco how some doubts about the present comes from thoughts anchored in the past. His belief that he did not deserve to live in his father’s house was linked to past episodes in which he perceived himself as selfish and ungrateful. Validating his desire for independence on several occasions lead to an explicit request to his father to move. Fortunately, the request was accepted by his parents with little resistance and Marco lived alone for a few months. His relationship with his parents changed enormously as did his misophonia, which was no longer present or of mild intensity when he returned for family lunches. He felt that his desire for autonomy could be satisfied, and he saw his parents again with more pleasure and when he wanted to. At the same time, not only in family but also in friendship relationships, we worked on the recognition of his own needs and the ability to put limits on the demands of others. In carrying out these exercises, Marco realized that it was not easy for him to identify his own desires because he was conflicted by the need to be ‘good’. By sharing how he functioned, he noticed that misophonia was often triggered when his friends ate food that he desired but considered unhealthy and denied himself in order to emerge as the wiser and more righteous one. Understanding this mechanism made it possible to read the presence of the sounds in a different way, realizing that the sense of constriction was often linked to his internal mechanism.

Two similar episodes that happened a few days apart were interesting.

P: On Monday I went to the cinema with a guy I just met, I ate what I wanted before going in, he was nice, he even drove without letting me take the car. He was complaining about some people eating in the hall. I was surprised because I had no misophonia! I wondered why and realized that I was happy and satisfied with the day, I had done what I wanted.

T: Well I’m glad! When your needs are met and you know you have chosen, you are more quiet.

P: Yes exactly, as we said. In fact, listen… on Saturday, when I went back to the cinema, I noticed that I was annoyed by some girls who were eating chips… so I asked myself how I felt and what was happening to me… before going to the cinema I had been persuaded by friends to get a sandwich that I didn’t like and I hadn’t taken the chips. I was hungry and those girls were eating what I wanted but was denying myself for the reasons we know. After doing this I felt better and was able to concentrate on the film. However, I couldn’t bring myself to buy the chips… partly because of the unhealthy food issue, partly because I still think eating at the cinema is rude… (laughs).

Another important step was to realize that it was not his responsibility to meet his parents’ needs in order to see them happy and consequently have a better self-perception. Marco was able to tell himself that ‘mum is always dissatisfied and fighting with the world, I realized that it is not my problem and that I am not the cause of this. The same for my father, I see him sad but he is the one who decided for this life, he cannot take it out on us sons’.

Working on interpersonal cycles with exposure exercises, Marco realized how his tendency to complacency and submission contributed to their activation. By eliminating these tendencies, e.g., with his father for work requests, he realized that when he was able to say ‘no’, he did not feel annoyed by the sounds his father made. By working on these dynamics, Marco realized that misophonia did not arise when he perceived that his demands were taken into account, accepted, and listened to. The improvements on misophonia are remarkable, but the work with Marco is not finished. The awareness of his functioning has paved the way for new therapeutic goals. Work is now focusing on the final stages of MIT treatment with the construction of a new self-narrative and the promotion of advanced mastery strategies, consisting of knowing how to voluntarily use learned psychological knowledge to cope with emotional distress, manage interpersonal conflicts, realize one’s own desires, and cooperate appropriately with others.

## 3. Results

The questionnaires were resubmitted after approximately two years of the treatment. The score obtained from the A-MISO-S was 8, a mild grade. The BDI-II reported no depressive symptoms (score 3) and the STAI-Y a medium-low level of both state-anxiety (score 34) and trait-anxiety (score 35). The SCID-II confirms the personality profile, as frequently happens, because patients are more aware of their own functioning.

Table 1 and Table 2 report the results at the beginning and after two years of the treatment.

Marco went back to live with his parents after returning from his work experience abroad. He no longer had angry outbursts and did not wear earphones at the dinner table. He recognised moments of stress and duress and shifted his attention to his internal dynamics or different stimuli. Misophonia, in the presence of friends and acquaintances, was absent. He understood and increasingly saw that constricting situations were due to active interpersonal patterns. He has increased his ability to stop and identify his needs in order to give them a voice in his relationship with others.

Therapy continued to solidify a different self-image as some work failures, such as the one abroad, and increased his sense of inadequacy and confusion about his life goals. He had mood swings at times, but he generally maintained an optimistic outlook; he paid more attention to his appearance by working out at the gym, and he dedicated himself to finding new job opportunities. As soon as economically possible, he will move back to living alone.

## 4. Discussion

The case study described and emphasised the importance of an accurate assessment to evaluate the presence of co-morbidity with other disorders and to identify interpersonal maladaptive schemas. 

Despite the obvious limitations of single case studies, these have the advantage of going into detail about the thoughts and emotions patients have in relation to trigger events. The attribution of intentionality to the actions of others is an element that often recurs, associated with viewing others as disrespectful, prevaricating, or rude [14,18,19,22].

Another element to consider is the image that patients have of themselves in their relationships with others; they often feel wrong, inadequate, vulnerable, constrained, or inferior [18,23]. Furthermore, the importance of interpersonal cycles is evident in the dynamic, also revealed in this case study, whereby if the other person does something for the patient, taking care of him/her, or if the patient thinks that he or she has hurt the other person, misophonia does not occur or decreases [30]. Interpersonal schemas originate from the child’s first attachment relationships with care givers. A hostile and unsupportive environment can condition the child’s way of relating to the world, hindering the achievement of his or her goals. An in-depth analysis of attachment styles and communication patterns within the family environment may be of interest in the study of misophonia in order to understand which particular conditions contribute to the onset of the interpersonal schemas underlying the disorder (e.g., parents separated at home, high conflict, aggression, or excessive control). In addition, misophonia often arises in late childhood and adolescence [3,12,36,37] when the individual, as the schemas become established, begins to have his or her own view of interpersonal relationships and begins to be increasingly autonomous from care givers. The details of the case presented and the results of the treatment point us to a framing of misophonia in the field of psychiatric disorders rather than hearing disorders. Further investigation is certainly necessary.

## 5. Conclusions

We think that emotional regulation and attention shifting techniques are useful for the treatment of misophonia but not sufficient. More specific work on managing relationships with others and thus on interpersonal patterns is necessary to diminish or eliminate symptoms. On this basis, we believe that MIT can be an effective therapy for the treatment of misophonia because, in addition to working on interpersonal schemas and continuous modulation and attention in the therapeutic relationship, it adds experiential techniques useful for managing arousal in the presence of misophonic stimuli.

## Figures and Tables

**Table 1 brainsci-14-00717-t001:** Results at the beginning of the treatment for Amsterdam Misophonia Scale (A-MISO-S), Structured Clinical Interview for DSM-IV Personality Disorders (SCID-II), Beck Depression Inventory-II (BDI-II), and State-Trait Anxiety Inventory (STAI-Y). Obsessive-Compulsive Personality Disorder (OCPD). Avoidant Personality Disorder (AvPD).

A-MISO-S Score	SCID-II	BDI-II Score	STAI-Y State-Anxiety Trait-Anxiety Score
16	OCPD	12	46
	AvPD		49

**Table 2 brainsci-14-00717-t002:** Results after two years of the treatment for Amsterdam Misophonia Scale (A-MISO-S), Structured Clinical Interview for DSM-IV Personality Disorders (SCID-II), Beck Depression Inventory-II (BDI-II), and State-Trait Anxiety Inventory (STAI-Y). Obsessive-Compulsive Personality Disorder (OCPD). Avoidant Personality Disorder (AvPD).

A-MISO-S Score	SCID-II	BDI-II Score	STAI-Y State-Anxiety Trait-Anxiety Score
8	OCPD	3	34
	AvPD		35

## Data Availability

The original contributions presented in the study are included in the article, further inquiries can be directed to the corresponding author.

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
