# Peer review of "Metacognitive Interpersonal Therapy for Misophonia: A Single-Case Study"

_brainsci, 2024, doi:10.3390/brainsci14070717_

Round 1
Reviewer 1 Report
Comments and Suggestions for Authors
General comment:
English is in need of meticulous revision
Introduction:
The authors should mention the relation between hyperacusis and mesophonia since the symptoms of their patient are triggered in any case by his hyperacusis.
The rationale and aim of the work are not mentioned
Where are the research questions?
Materials and Method:
In the section of "assessment and instruments, the authors need to clarify how scores of different instruments are calculated and the classification of morbidity of each scale.
The section of “Shared formulation of functioning” needs to be rearranged in a more organized structure.
The objective and aim of each step of treatment should be mentioned
The objective of the “Promoting change” section should be mentioned
Results:
The authors have to mention the results of the scales used to assess their case before and after MIT to show the improvement that was achieved by their patient. A table with this data will be adequate.
Discussion:
Almost there is no discussion at all!!!!
In this study, the authors related their patient’s misophonia to be related to the sense of guilt and disgust. The domestic unhealthy environment played a critical role in his illness. This deserves proper discussion.
Comments on the Quality of English LanguageNeed revision. Many statement are difficult to understand.
Author Response
Thank you for the careful review of our manuscript and insightful comments.
We hope our efforts are in line with your expectations.
Comment 1
English is in need of meticulous revision
Response 1
Thanks for your valuable suggestion. We did a significant revision of English. If it's not sufficient we will have ask our manuscript revision by rapid English editing of MDPI.
Comment 2
-
The authors should mention the relation between hyperacusis and mesophonia since the symptoms of their patient are triggered in any case by his hyperacusis.
Response 2
Thank you for your valuable comment. We think that there is a misunderstanding because our patient had normal hearing and no hyperacusis as evaluated with normal Loudness discomfort Level (LDL). We have corrected the sentence at the end of assessment and instruments:
Hyperacusis was assessed excluded testing with the loudness discomfort levels (LDLs) at the frequencies of 0.25, 0.5, 1, 2, 4 and 8 kHz. The patient had normal hearing and normal LDLs.
Comment 3
The rationale and aim of the work are not mentioned
Where are the research questions?
Response 3
We deeply appreciate your comments. Accordingly, we have clarified the purpose of the study in the final part of the introduction. The study aims to highlight the importance of interpersonal patterns in maintaining the disorder.
The purpose of our study is to detail the MIT treatment of a patient with misophonia in order to highlight the importance of interpersonal schemas in maintaining the disorder. To our knowledge, this is the first study of misophonia successfully treated with MIT.
Comment 4
In the section of "assessment and instruments, the authors need to clarify how scores of different instruments are calculated and the classification of morbidity of each scale.
Response 4
Thank you for your valuable comment. We have added a more detailed description of the questionnaires used.
Several questionnaires were administered at the beginning of treatment. The diagnosis of misophonia was made by clinical interview based on the criteria described by Schröder and colleagues [3] and with the Amsterdam Misophonia Scale (A-MISO-S). [3]. Structured clinical interview for DSM-IV personality disorders (SCID-II)[33], Beck depression inventory-II (BDI-II) [34] and State-trait anxiety inventory (form-Y) (STAI-Y) [35] were also administered at the beginning of treatment.The A-MISO-S is a semi-structured interview which is not validated. On 6-item scale (range 0–24) patients were asked about the (1) time they spend on misophonia, (2) interference with social functioning, (3) level of anger, (4) resistance against the impulse, (5) control they had over their thoughts and anger, and (6) time they spend avoiding misophonic situations. Scores from 0 to 4 are considered subclinical misophonic symptoms, 5–9 mild, 10–14 moderate, 15–19 severe and 20–24 extreme. The SCID-II is a structured clinical interview that assesses the full range of PD traits found in DSM IV PD. The interview was administered by the treating clinician before the beginning of any physical or psychological treatment.The BDI-II is a 21-item measure assessing depression over the previous 2 weeks. Higher scores suggest a high level of depression. The cutoff used are the following: 0–13 corresponded to minimal depression, 14–19 to mild depression, 20–28 to moderate depression and 29–63 to severe depression. The STAI-Y is a self-report instrument measuring state-anxiety (anxiety about an event) and trait-anxiety (anxiety level as a stable characteristic). All items were rated on a 4-point Likert Scores range from 20 to 80. Higher scores have a correlation with a higher level of anxiety.
Comment 5
The section of “Shared formulation of functioning” needs to be rearranged in a more organized structure.
The objective and aim of each step of treatment should be mentioned
Response 5
We deeply appreciate your comments. Accordingly, we have better clarified the main steps of the collection of narrative episodes, the connected autobiographical memories and the reconstruction of the schemes.
As indicated by MIT, the first part of the treatment focused on the construction of a shared formulation of the patient's functioning. In the first meetings, exercises of exploration and observation of the internal state were assigned to improve narrative and self-reflective skills.
Marco was asked to remove the earphones at trigger moments related to misophonia to check which thoughts, emotions, physical sensations and action tendencies he was experiencing. Attentional techniques to be used in case of excessive discomfort were described and explained. Marco was already familiar with mindfulness and this helped the process. In addition, he was given descriptive emotion cards to help the patient name his feelings.
The first part of the treatment was used to collect narrative episodes, i.e. detailed autobiographical memories that are well located in time and space. The focus is then placed on the details of these episodes to search for feelings, ideas, and motivations for actions. The observation and narration of the episodes led to the identification of several interesting points. Anger, typical of misophonia, emerged because Marco felt forced by others, not only to witness unpleasant sounds, but because others ate unhealthy food. This last observation was related to the feeling of disgust, of a moral kind, which also emerged in another observation. Marco felt more misophonia-related distress towards people who did not help tidy up the table and whom he judged to be rude, with no respect for others.
The exercises also helped to identify problems within the family. At the table, quarrels and discontent were the order of the day and Marco suffered especially for his mother, described as complaining and discontented. His mother often refused to eat with them when she was angry and aroused feelings of anger and guilt in Marco. In the past, there was aggressive behaviour when parents argued, with the father breaking objects and the mother damaging her own person, as a demonstrative act of her suffering.
On a bodily level, Marco's physical sensations were characterized by stomach tension and muscular rigidity that could result in aggressive behaviour that Marco later regretted.
T.: do you feel guilty after these behaviours?
P.: yes very much...not expressing anger is a quality, I feel better than others when I succeed. I feel that I disturb others if I am angry.
T.: Why do you have this thought? Did someone tell you that in the past?
P.: When I was a child I was lively, I did damages, I even beat up a friend at school. My grandparents told me how I used to disturb everyone. And finally I realised that I had to be calm and not be a nuisance, even in class... also because I wasn't really bright (laughs)... but at least I was appreciated for my behaviour. I always felt a bit of a loser at school, less handsome, less intelligent....
With the therapist Marco began to see how misophonia was the only way to express anger because, from his point of view, his complaints were correct and the others were in the wrong.
After analyzing the misophonia episodes, he was asked to bring back other episodes related to the emotion of anger in therapy to see what interpersonal patterns might be activated. The investigation was not easy because, outside of misophonia, for the reasons described above, it was difficult for Marco to tell himself that he was feeling that emotion.
It was possible to identify moments of strong stress related to his father's requests for help at work. Marco did not want often to help him, but did so because he would otherwise feel selfish. The feeling of guilt also came back when his mother showed signs of impatience or sadness.
A healthy desire to live alone emerged during the interviews. The family owns a house that was sometimes used by the father because it was closer to work and because of the bad relationship with his wife. Some meetings were dedicated to the possibility of using this house, but Marco found it very difficult to ask, he felt he did not deserve this possibility. During the session, the therapist led Marco to evoke autobiographical memories related to the current situation, in order to show the patient the existence of recurrences in the way of relating.
T: Does this belief that you don't deserve things remind you of something from the past? Where could you have heard it?
P: When I was a child, I never had fashionable clothes and I was very ashamed. My mother wouldn't buy them for me not to spend money. I was angry but then I stopped asking... I don't know then at some point it felt right. Once, however, I had asked for a game and she bought it for me... I wanted it so much, I was happy, but then my mother... (she turns dark)
T: What's happening? I can see sadness on your face...
P: yes... I remember that I saw her suffering... it wasn't a gesture made in joy, I felt that it was wrong, that the shopping was damaging the family, that I was a bad child... instead I had to be a good child, I didn't have to cause problems.
T: And how did you feel towards your mother?
P: I felt guilty. Both she and dad always said they stayed together for us children and did things for us. I felt... I don't know... ungrateful, if I only thought of myself. It's dawning on me that my mother also used to complain a lot when she had to take me to football school, she made me feel it was weighing on me...
The informations obtained through the narrative episodes and autobiographical memories allowed the reconstruction of interpersonal schemas. The desire for autonomy was hindered by the perception of himself as selfish and ungrateful if he took care of his own needs, and he felt guilt for this. The perception of the other was of a person who could be hurt by his desires or needs. In this case he felt the relationship threatened. Sometimes, however, the feeling of anger was present when he perceived the other as controlling and domineering. In this case Marco felt that the other wanted to subjugate him and his self-image as a person capable and deserving of autonomy was activated. In this case the healthy part emerged, as it is called in MIT. Perceiving the other as dominant also activated the motivational schema of social rank. This emerged to counter the self-image, not only as inferior because submissive, but also inadequate and incapable. High morality and the need to follow certain rules were used in order to perceive oneself as better than others and wise. The dysfunctional coping strategies related to the schemas were perfectionism, the use of high morality, and avoidance.
The reconstruction of the patterns was accepted by Marco with great interest and openness. Marco agreed with the reported reflections and this allowed the transition to the next phase of therapy concerning change.
Comment 6
The objective of the “Promoting change” section should be mentioned
Response 6
Thank you for your valuable comment. We added the object in the first part of the paragraph.
2.3.2. Promothing change
At the promothing change stage, clients are helped to take a critical distance from their schemas to build new ways of thinking and feeling in order to implement more adaptive behaviours.
Comment 7
The authors have to mention the results of the scales used to assess their case before and after MIT to show the improvement that was achieved by their patient. A table with this data will be adequate.
Response 7
We deeply appreciate your comments. Accordingly, we have added 2 tables in the results.
Tables 1 and 2 reported the results at the beginning and after two years of treatment.
|
A-MISO-S score |
SCID-II |
BDI-II score |
STAI-Y State-Anxiety Trait-Anxiety score |
|
16 |
OCPD AvPD |
12 |
46 49 |
Table 1. Results at the beginning of treatment for Amsterdam Misophonia Scale (A-MISO-S), Structured Clinical Interview for DSM-IV Personality Disorders (SCID-II), Beck Depression Inventory-II (BDI-II) and State-Trait Anxiety Inventory (STAI-Y). Obsessive-Compulsive Personality Disorder (OCPD). Avoidant Personality Disorder (AvPD)
|
A-MISO-S score |
SCID-II |
BDI-II score |
STAI-Y State-Anxiety Trait-Anxiety score |
|
8 |
OCPD AvPD |
3 |
34 35 |
Table 2. Results after two years of treatment for Amsterdam Misophonia Scale (A-MISO-S), Structured Clinical Interview for DSM-IV Personality Disorders (SCID-II), Beck Depression Inventory-II (BDI-II) and State-Trait Anxiety Inventory (STAI-Y). Obsessive-Compulsive Personality Disorder (OCPD). Avoidant Personality Disorder (AvPD)
Comment 8
Almost there is no discussion at all!!!!
In this study, the authors related their patient’s misophonia to be related to the sense of guilt and disgust. The domestic unhealthy environment played a critical role in his illness. This deserves proper discussion.
Response 8
Thank you for your valuable comment. We have enriched the discussion with other elements and better explained the family impact. Guilt or disgust are not directly linked to misophonia, but are emotions present in interpersonal cycles. The unhealthy environment contributes to the creation of maladaptive interpersonal patterns.
The case study described, emphasizes the importance of an accurate assessment to evaluate the presence of co-morbidity with other disorders and to identify interpersonal maladaptive schemas. Despite the obvious limitations of single case studies, these have the advantage of going into detail about the thoughts and emotions patients have in relation to trigger events. The attribution of intentionality to the actions of others is an element that often recurs, associated with viewing others as disrespectful, prevaricating, rude [13,17,18, 2114,18,19,22]. Another element to consider is the image patients have of themselves in their relationships with others; they often feel wrong, inadequate, vulnerable, constrained or inferior [17,2218,23]. Furthermore, the importance of interpersonal cycles is evident in the dynamic, also revealed in this case study, whereby if the other person does something for the patient, taking care of him/her, or if the patient thinks that he or she has hurted the other person, misophonia does not occur or decreases [2930]. Interpersonal schemas originate from the child's first attachment relationships with care givers. A hostile and unsupportive environment can condition the child's way of relating to the world, hindering the achievement of his or her goals. An in-depth analysis of attachment styles and communication patterns within the family environment may be of interest in the study of misophonia in order to understand which particular conditions contribute to the onset of the interpersonal schemas underlying the disorder (e.g. parents separated at home, high conflict, aggression, excessive control). In addition, misophonia often arises in late childhood and adolescence [3, 12, 36, 37] when the individual, as the schemas become established, begins to have his or her own view of interpersonal relationships and begins to be increasingly autonomous from care givers. The details of the case presented and the results of the treatment point us to a framing of misophonia in the field of psychiatric disorders rather than hearing disorders. Further investigation is certainly necessary.
Reviewer 2 Report
Comments and Suggestions for Authors
General comments
The manuscript is a case report on Metacognitive Interpersonal Therapy for Misophonia. The manuscript is clearly written and very informative. I just wonder if some information about the patient could not be moved to the supplementary material in order to shorten the manuscript and just focus on the results of the therapy.
The manuscript is intended for special issue on tinnitus. However, it is unclear how it is related to tinnitus and specifically to the topic of special issue. I suggest that authors add some rationale for this in the introduction section, i.e. how misophonia is related to tinnitus.
My more detailed comments are listed below.
Abstract
L24 – “increased” – should not it be “increase”?
Introduction
I would expect some more connection with the tinnitus since the manuscript is intended to the special issue on the tinnitus.
I suggest to provide some more information on how Metacognitive Interpersonal Therapy actually look like.
Please formulate more directly the aim of the study.
Results
Maybe it would be possible to add a figure showing the results? Maybe it would be possible to make some comparison of Metacognitive Interpersonal Therapy with some other methods, or maybe with the results of the patient before or after the therapy?
References
Please correct double numbers in the references.
Author Response
Comment 1
The manuscript is a case report on Metacognitive Interpersonal Therapy for Misophonia. The manuscript is clearly written and very informative. I just wonder if some information about the patient could not be moved to the supplementary material in order to shorten the manuscript and just focus on the results of the therapy.
Response 1
Thank you for the careful review of our manuscript and insightful comment but we think that patient data and informations are an integral part of the study to understand how it works.
Comment 2
The manuscript is intended for special issue on tinnitus. However, it is unclear how it is related to tinnitus and specifically to the topic of special issue. I suggest that authors add some rationale for this in the introduction section, i.e. how misophonia is related to tinnitus.
Response 2
Thank you for your valuable comment. We have added the relationship between misophonia and tinnitus in the introduction, as you suggested.
Tinnitus is a phantom auditory perception and, based on the neurophysiological model of Jastreboff, the neuronal networks involved in tinnitus and misophonia are identical [6]. The auditory system is needed for perception of tinnitus and misophonia but the limbic and autonomic nervous systems are the main systems responsible for negative tinnitus and misophonia evoked reactions. Conditioned reflexes explains why there are problems with tinnitus or misophonic triggers. There are two paths in network processing tinnitus signal and activity evoked by bothersome sounds. A conscious path which involves cognitive processing of the signal, and which is dominant at the initial stages of tinnitus or misophonia. A subconscious path, governed by principles of conditioned reflexes appears to be dominant in chronic tinnitus or misophonia.
Comment 3
Abstract
L24 – “increased” – should not it be “increase”?
Response 3
Thank you, we have made the correction requested.
Comment 4
I suggest to provide some more information on how Metacognitive Interpersonal Therapy actually look like.
Response 4
We deeply appreciate your comments. Accordingly, we have added in the paragraph 2.3 (course of treatment) more details in the passages that describe the MIT, better clarifying the object and purpose of the various phases of the treatment.
2.3.1. Shared formulation of functioning
As indicated by MIT, the first part of the treatment focused on the construction of a shared formulation of the patient's functioning. In the first meetings, exercises of exploration and observation of the internal state were assigned to improve narrative and self-reflective skills.
Marco was asked to remove the earphones at trigger moments related to misophonia to check which thoughts, emotions, physical sensations and action tendencies he was experiencing. Attentional techniques to be used in case of excessive discomfort were described and explained. Marco was already familiar with mindfulness and this helped the process. In addition, he was given descriptive emotion cards to help the patient name his feelings.
The first part of the treatment was used to collect narrative episodes, i.e. detailed autobiographical memories that are well located in time and space. The focus is then placed on the details of these episodes to search for feelings, ideas, and motivations for actions. The observation and narration of the episodes led to the identification of several interesting points. Anger, typical of misophonia, emerged because Marco felt forced by others, not only to witness unpleasant sounds, but because others ate unhealthy food. This last observation was related to the feeling of disgust, of a moral kind, which also emerged in another observation. Marco felt more misophonia-related distress towards people who did not help tidy up the table and whom he judged to be rude, with no respect for others.
The exercises also helped to identify problems within the family. At the table, quarrels and discontent were the order of the day and Marco suffered especially for his mother, described as complaining and discontented. His mother often refused to eat with them when she was angry and aroused feelings of anger and guilt in Marco. In the past, there was aggressive behaviour when parents argued, with the father breaking objects and the mother damaging her own person, as a demonstrative act of her suffering.
On a bodily level, Marco's physical sensations were characterized by stomach tension and muscular rigidity that could result in aggressive behaviour that Marco later regretted.
With the therapist Marco began to see how misophonia was the only way to express anger because, from his point of view, his complaints were correct and the others were in the wrong.
After analyzing the misophonia episodes, he was asked to bring back other episodes related to the emotion of anger in therapy to see what interpersonal patterns might be activated. The investigation was not easy because, outside of misophonia, for the reasons described above, it was difficult for Marco to tell himself that he was feeling that emotion.
It was possible to identify moments of strong stress related to his father's requests for help at work. Marco did not want often to help him, but did so because he would otherwise feel selfish. The feeling of guilt also came back when his mother showed signs of impatience or sadness.
A healthy desire to live alone emerged during the interviews. The family owns a house that was sometimes used by the father because it was closer to work and because of the bad relationship with his wife. Some meetings were dedicated to the possibility of using this house, but Marco found it very difficult to ask, he felt he did not deserve this possibility. During the session, the therapist led Marco to evoke autobiographical memories related to the current situation, in order to show the patient the existence of recurrences in the way of relating.
T: Does this belief that you don't deserve things remind you of something from the past? Where could you have heard it?
P: When I was a child, I never had fashionable clothes and I was very ashamed. My mother wouldn't buy them for me not to spend money. I was angry but then I stopped asking... I don't know then at some point it felt right. Once, however, I had asked for a game and she bought it for me... I wanted it so much, I was happy, but then my mother... (she turns dark)
T: What's happening? I can see sadness on your face...
P: yes... I remember that I saw her suffering... it wasn't a gesture made in joy, I felt that it was wrong, that the shopping was damaging the family, that I was a bad child... instead I had to be a good child, I didn't have to cause problems.
T: And how did you feel towards your mother?
P: I felt guilty. Both she and dad always said they stayed together for us children and did things for us. I felt... I don't know... ungrateful, if I only thought of myself. It's dawning on me that my mother also used to complain a lot when she had to take me to football school, she made me feel it was weighing on me...
The informations obtained through the narrative episodes and autobiographical memories allowed the reconstruction of interpersonal schemas. The desire for autonomy was hindered by the perception of himself as selfish and ungrateful if he took care of his own needs, and he felt guilt for this. The perception of the other was of a person who could be hurt by his desires or needs. In this case he felt the relationship threatened. Sometimes, however, the feeling of anger was present when he perceived the other as controlling and domineering. In this case Marco felt that the other wanted to subjugate him and his self-image as a person capable and deserving of autonomy was activated. In this case the healthy part emerged, as it is called in MIT. Perceiving the other as dominant also activated the motivational schema of social rank. This emerged to counter the self-image, not only as inferior because submissive, but also inadequate and incapable. High morality and the need to follow certain rules were used in order to perceive oneself as better than others and wise. The dysfunctional coping strategies related to the schemas were perfectionism, the use of high morality, and avoidance.
The reconstruction of the patterns was accepted by Marco with great interest and openness. Marco agreed with the reported reflections and this allowed the transition to the next phase of therapy concerning change.
2.3.2. Change promothing
At the change promothing stage, clients are helped to take a critical distance from their schemas to build new ways of thinking and feeling in order to implement more adaptive behaviours.
A first step towards differentiation, that is taking a critical distance from one's schemas, was obtained with the use of mindfulness and guided imagination to remind Marco how some doubts about the present comes from thoughts anchored in the past. In fact, as we saw earlier, his belief that he did not deserve to live in his father's house was linked to past episodes in which he perceived himself as selfish and ungrateful. Validating his desire for independence on several occasions lead to an explicit request to his father to move. Fortunately, the request was accepted by his parents with little resistance and Marco lived alone for a few months. His relationship with his parents changed enormously as did his misophonia, which was no longer present or of mild intensity when he returned for family lunches. He felt that his desire for autonomy could be satisfied and he saw his parents again with more pleasure and when he wanted to. At the same time, not only in family but also in friendship relationships, we worked on the recognition of his own needs and the ability to put limits on the demands of others. In carrying out these exercises Marco realized that it was not easy for him to identify his own desires because he was conflicted by the need to be ‘good’. By sharing how he functioned, he noticed that misophonia was often triggered when his friends ate food that he desired but considered unhealthy and denied himself in order to emerge as the wiser and more righteous one. Understanding this mechanism made it possible to read the presence of the sounds in a different way, realizing that the sense of constriction was often linked to his internal mechanism.
Two similar episodes that happened a few days apart were interesting.
P: On Monday I went to the cinema with a guy I just met, I ate what I wanted before going in, he was nice, he even drove without letting me take the car. He was complaining about some people eating in the hall. I was surprised because I had no misophonia! I wondered why and realized that I was happy and satisfied with the day, I had done what I wanted.
T: well I'm glad! When your needs are met and you know you have chosen, you are more quiet.
P: yes exactly, as we said. In fact, listen... on Saturday, when I went back to the cinema, I noticed that I was annoyed by some girls who were eating chips... so I asked myself how I felt and what was happening to me... before going to the cinema I had been persuaded by friends to get a sandwich that I didn't like and I hadn't taken the chips. I was hungry and those girls were eating what I wanted but was denying myself for the reasons we know. After doing this I felt better and was able to concentrate on the film. However, I couldn't bring myself to buy the chips... partly because of the unhealthy food issue, partly because I still think eating at the cinema is rude... (laughs)
Another important step was to realize that it was not his responsibility to meet his parents' needs in order to see them happy and consequently have a better self-perception. Marco was able to tell himself ‘mum is always dissatisfied and fighting with the world, I realized that it is not my problem and that I am not the cause of this. The same for my father, I see him sad but he is the one who decided for this life, he cannot take it out on us sons’.
Working on interpersonal cycles with exposure exercises, Marco realized how his tendency to complacency and submission contributed to their activation. By eliminating these tendencies, e.g. with his father for work requests, he realized that when he was able to say ‘no’, he did not feel annoyed by the sounds his father made. By working on these dynamics, Marco realized that misophonia did not arise when he perceived that his demands were taken into account, accepted and listened to. The improvements on misophonia are remarkable but the work with Marco is not finished. The awareness of his functioning has paved the way for new therapeutic goals. Work is now focusing on the final stages of MIT treatment with the construction of a new self-narrative and the promotion of advanced mastery strategies, consisting of knowing how to voluntarily use learnt psychological knowledges to cope with emotional distress, manage interpersonal conflicts, realize one's own desires and cooperate appropriately with others.
Comment 5
Please formulate more directly the aim of the study.
Response 5
Thank you for your valuable comment. We have added the aim of the study at the end of the introduction:
The purpose of our study is to detail the MIT treatment of a patient with misophonia in order to highlight the importance of interpersonal schemas in maintaining the disorder. To our knowledge, this is the first study of misophonia successfully treated with MIT.
Comment 6
Maybe it would be possible to add a figure showing the results? Maybe it would be possible to make some comparison of Metacognitive Interpersonal Therapy with some other methods, or maybe with the results of the patient before or after the therapy?
Response 6
Thank you for your valuable comment. We have added 2 tables in the results showing the results of the patient before of after the therapy:
|
A-MISO-S score |
SCID-II |
BDI-II score |
STAI-Y State-Anxiety Trait-Anxiety score |
|
16 |
OCPD AvPD |
12 |
46 49 |
Table 1. Results at the beginning of treatment for Amsterdam Misophonia Scale (A-MISO-S), Structured Clinical Interview for DSM-IV Personality Disorders (SCID-II), Beck Depression Inventory-II (BDI-II) and State-Trait Anxiety Inventory (STAI-Y). Obsessive-Compulsive Personality Disorder (OCPD). Avoidant Personality Disorder (AvPD)
|
A-MISO-S score |
SCID-II |
BDI-II score |
STAI-Y State-Anxiety Trait-Anxiety score |
|
8 |
OCPD AvPD |
3 |
34 35 |
Table 2. Results after two years of treatment for Amsterdam Misophonia Scale (A-MISO-S), Structured Clinical Interview for DSM-IV Personality Disorders (SCID-II), Beck Depression Inventory-II (BDI-II) and State-Trait Anxiety Inventory (STAI-Y). Obsessive-Compulsive Personality Disorder (OCPD). Avoidant Personality Disorder (AvPD)
Comment 7
References
-
Please correct double numbers in the references.
Response 7
Thank you for your valuable comment. We have corrected double numbers in the references.
We hope our efforts are in line with your expectations.
Reviewer 3 Report
Comments and Suggestions for Authors
In this manuscript from Natalini et al., the authors have presented the case report of a young patient who struggled with misophonia. The patient underwent metacognitive interpersonal therapy. After about 2 years of therapy, the patient showed improvement in misophonia measure and also of BD-II, STAI-Y scores. I only have two minor comments:
1) The therapist-patient conversation presented in lines 179-186, 204-219, 259-273 can be moved to a supplementary file. It's not clear whether it adds any additional value being in the main text.
2) The writing style of the manuscript is very different from what's expected of a research article. The style is narrative and is similar to story telling. Although not absolutely necessary, the manuscript can be improved by changing the style.
Author Response
Comment 1
The therapist-patient conversation presented in lines 179-186, 204-219, 259-273 can be moved to a supplementary file. It's not clear whether it adds any additional value being in the main text.
Response 2
Thank you for the careful review of our manuscript and insightful comments. We agree with your comment and have accordingly removed the first exchange between the patient and the therapist because the dialogue was not fundamental. We think that the other conversations, however, are important to highlight some steps of the treatment.
Comment 2
The writing style of the manuscript is very different from what's expected of a research article. The style is narrative and is similar to story telling. Although not absolutely necessary, the manuscript can be improved by changing the style.
Response 2
Thank you for your valuable comment. The writing style is different because in the psychological field it is often necessary to describe the patient's internal and external world to clarify his functioning. However, we have made some changes (e.g. therapeutic steps are now more explicit, we have eliminated a dialogue and inserted 2 tables in the results as reported in red in the file attached)
We hope our efforts are in line with your expectations.
Round 2
Reviewer 1 Report
Comments and Suggestions for Authors
Thanks for your meticulous revision, the article is much better now.